# Two-dimensional higher-order Weyl semimetals

Lizhou Liu,[1] Qing-Feng Sun,[2, 3, *] and Ying-Tao Zhang[1, †]

[1]*College of Physics, Hebei Normal University, Shijiazhuang 050024, China*
[2]*International Center for Quantum Materials, School of Physics, Peking University, Beijing 100871, China*
[3]*Hefei National Laboratory, Hefei 230088, China*
(Dated: December 5, 2025)

We propose a theoretical scheme to realize two-dimensional higher-order Weyl semimetals using a trilayer topological film coupled with a $d$-wave altermagnet. Our results show that the trilayer topological film exhibits two-dimensional Weyl semimetal characteristics with helical edge states. Notably, the Weyl points are located at four high-symmetry points in the Brillouin zone, and the topology of symmetric subspaces governs the formation of these Weyl points and edge states. Upon introducing a $d$-wave altermagnet oriented along the $z$-direction, gaps open in the helical edge states while preserving two Weyl points, leading to the realization of two-dimensional higher-order Weyl semimetals hosting topological corner states. The nonzero winding number in the subspace along the high-symmetry line serves as a topological invariant characterizing these corner states, and the other subspace Hamiltonian confirms the existence of the Weyl points. Finally, a topological phase diagram provides a complete topological description of the system.

## I. INTRODUCTION

In recent years, the study of topological phases has expanded significantly, uncovering novel properties across a wide range of systems [1, 2]. Beyond the traditional bulk-boundary correspondence observed in topological insulators [3–10] and semimetals [11–14], the emergence of higher-order topological phases has marked a paradigm shift. A $d$-dimensional $n$th-order topological phase is characterized by gapless boundary modes living on $(d − n)$-dimensional boundaries rather than on the conventional $(d − 1)$-dimensional ones. In particular, in two dimensions a second-order topological phase has all one-dimensional edges gapped, while zero-dimensional corner states appear inside the edge excitation gap. Initially proposed for insulators, these higher-order topological phases exhibit distinctive bulk-boundary relations, including corner states in two- or three-dimensional systems and hinge states in three-dimensional systems [15–25]. A major research focus has been the induction of higher-order corner states by breaking the edge states of two-dimensional (2D) topological insulators [26–34].

Building on these 2D corner states, second-order topological semimetal states with hinge states, such as Weyl semimetals, Dirac semimetals, and nodal ring semimetals, have been predicted in bulk-closed three-dimensional systems [35–48]. The dimension reduction may produce unique physical properties, such as parity anomaly in $(2 + 1)$-dimensional (space-time) quantum field theory [49–52], giant Berry curvature dipole [53, 54], and topological quantum criticality [55]. Moreover, in contrast to hinge states, corner states in higher-order topological systems exhibit enhanced spatial localization and greater tunability [56], rendering them promising candidates for novel quantum functionalities. Incorporating such localized zero-dimensional modes into semimetal systems, which are characterized by gapless bulk, may give rise to unconventional boundary phenomena. Despite its theoretical appeal, the realization and characterization of higher-order topology—specifically, the presence of corner states—in two-dimensional semimetals remain largely unaddressed. A systematic exploration of this regime is therefore of fundamental importance, as it may uncover new classes of gapless higher-order phases and broaden the conceptual framework of topological quantum matter beyond the conventional gapped paradigm.

Recently, altermagnetism, distinct from both ferromagnetism and antiferromagnetism, has been proposed within the framework of spin group theory [57–59]. In altermagnetic systems, sublattices with opposite spins are not related by spatial inversion or fractional translation operations but are instead connected by rotation or mirror symmetry operations. This unique symmetry constraint gives rise to non-relativistic, anisotropic spin splitting in the Brillouin zone. As a result, altermagnets-combining real-space antiferromagnetic order with momentum-space anisotropic spin splitting, present a promising platform for realizing 2D higher-order semimetals.

In this work, we theoretically propose a pioneering scheme to realize 2D higher-order Weyl semimetals with corner states, utilizing trilayer topological films and $d$-wave altermagnets. As illustrated in Fig. 1(a), the 2D system hosts helical edge states and four bulk Weyl points at high symmetry points. Introducing a $d$-wave altermagnet along the $z$-direction opens a gap in the helical edge states while preserving two bulk Weyl points. The emergence of corner states within the edge gaps signifies the realization of 2D higher-order Weyl semimetals, as shown in Fig. 1(b). The topological invariants of the symmetric subspace characterise the topological nature of the system. Finally, a topological phase diagram is determined.

* Correspondence author: sunqf@pku.edu.cn
† Correspondence author: zhangyt@mail.hebtu.edu.cn

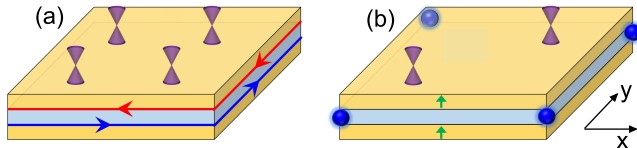

FIG. 1. (a) Schematic of a trilayer topological film. The films are represented in yellow and light blue layer, respectively. Red and blue lines indicate helical edge states. (b) Schematic of a trilayer film with $d$-wave altermagnet. The green arrows indicate the direction of the altermagnet parallel to the out-of-plane. The purple cones indicate the Weyl cones in the first Brillouin zone and the blue spheres indicate the corner states.

## II. SYSTEM HAMILTONIAN

We design the theoretical model based on the Hamiltonian form of the $Bi_2Se_3$ topological film [60–62], the Hamiltonian for the trilayer films as shown in Fig. 1 can be expressed as follows:

$$H(\boldsymbol{k}) = \upsilon_F(\sin k_y s_x - \sin k_x s_y)\rho_z + m(\boldsymbol{k})s_0\rho_x, \quad (1)$$

where $\boldsymbol{k} = (k_x, k_y)$, the first term corresponds to the kinetic energy, $\upsilon_F$ is the Fermi velocity, and the second term is the coupling term between the different layers, $m(\boldsymbol{k}) = m_0 + 4m_1 - 2m_1\cos k_x - 2m_1\cos k_y$, $m_0$ and $m_1$ are the hybridization gap and parabolic band components, respectively. Here $\rho_{x,z}$ denote the layer degrees of freedom, and the three degrees of freedom correspond to the top, middle, and bottom layers, the concrete form is

$$\rho_z = \begin{bmatrix} 1 & 0 & 0 \\ 0 & -1 & 0 \\ 0 & 0 & 1 \end{bmatrix}, \qquad \rho_x = \begin{bmatrix} 0 & 1 & 0 \\ 1 & 0 & 1 \\ 0 & 1 & 0 \end{bmatrix}. \quad (2)$$

Notably, the higher-order Weyl phase, along with the associated helical edge and corner states, arises only under the antisymmetric Fermi velocity configuration $(1, -1, 1)$. In contrast, alternative configurations such as $(-1, 1, 1)$ or $(1, 1, 1)$ fail to induce the necessary band inversions and do not support topologically protected boundary modes. The $(1, -1, 1)$ configuration, representing opposite Fermi velocity signs between the middle and outer layers, can in principle be realized via structural asymmetry or layer-selective gating in van der Waals trilayers. And $s_{x,y,z}$ in Eq. (1) is the Pauli matrices for the spin. The system is assumed to be half-filled with the Fermi level at zero energy.

The model presented in Eq. (1) extends the low-energy theory of $Bi_2Se_3$ thin films [60] to a trilayer configuration. It is formulated as a lattice model incorporating interlayer hybridization and spin-momentum locking and features idealized symmetries, such as chiral symmetry in specific momentum subspaces, that enable analytical topological classification. Although not derived from a $\mathbf{k} \cdot \mathbf{p}$ expansion around the $\Gamma$ point, and not intended

to quantitatively reproduce the full band structure of a real $Bi_2Se_3$ trilayer, the model provides a phenomenological framework for exploring higher-order topological mechanisms. In particular, the symmetric interlayer coupling facilitates the coexistence of gapless Weyl points and gapped helical edge states, enabling the emergence of corner states even without a full bulk gap. This construction thus offers a tunable platform for investigating second-order topology in two-dimensional Weyl systems.

The spin splitting of altermagnets exhibits various forms, including $p$-wave, $d$-wave, $g$-wave, and $i$-wave [59]. We employ $d$-wave spin splitting altermagnets, as follows

$$H_{AM} = J(\cos k_x - \cos k_y)s_z \otimes \begin{bmatrix} 1 & 0 & 0 \\ 0 & 0 & 0 \\ 0 & 0 & 1 \end{bmatrix}. \quad (3)$$

Here $s_z$ represents altermagnet in the $z$-direction, and only the top and bottom layers are affected by altermagnet, which can be induced by magnetic proximity effect [63].

Although time-reversal symmetry can be broken by conventional magnetic ordering, we deliberately employ a $d$-wave altermagnetic exchange field due to its momentum-dependent structure. This unique feature enables selective gap opening in the edge states while preserving the bulk Weyl points, thus realizing an ideal "gapless-bulk, gapped-boundary" topological phase. In contrast, conventional magnetic orders typically act more uniformly on both bulk and boundary states, making them less suitable for achieving such coexistence.

The $d$-wave altermagnetic term introduced in Eq. (3) is not postulated arbitrarily but can be microscopically justified from a perturbative self-energy correction induced by interfacial coupling between the Weyl semimetal and an altermagnet. Specifically, considering a tunneling strength $t$ and an altermagnet Hamiltonian $\widetilde{H}_{AM} = \widetilde{J}(\cos k_x - \cos k_y)s_z + \mu$, the leading-order self-energy correction takes the form

$$\Sigma^r_{k_x,k_y} \approx \frac{t^2}{\mu^2}\widetilde{J}(\cos k_x - \cos k_y)s_z,$$

which naturally leads to a momentum-dependent $d$-wave exchange structure consistent with Eq. (3). In addition, this form is constructed based on symmetry considerations and a theoretical understanding of spin-splitting patterns in altermagnetic systems. Notably, similar momentum-dependent $d$-wave exchange terms have been employed in recent theoretical studies [33, 64–66], which support the formal structure adopted in our model.

Our effective Hamiltonian $\tilde{H}_{AM} = J(\cos k_x - \cos k_y)\sigma_z$ is a phenomenological model designed to capture the essential feature of momentum-selective spin splitting. This form corresponds to the $B_{1g}$ irreducible representation in a tetragonal crystal and is consistent with the general symmetry-allowed terms discussed in Ref. [67]. In their classification, this term arises from a Néel-type altermagnetic order with inversion-symmetric orbitals located at Wyckoff positions compatible with $D_{2h}$ or $C_{2h}$

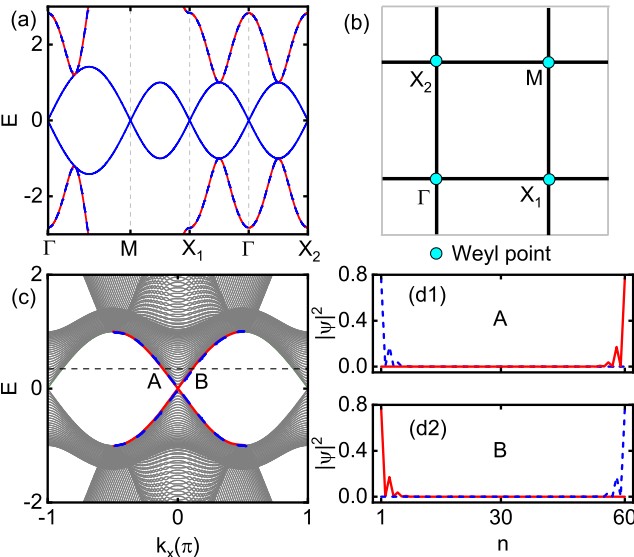

FIG. 2. (a) Bulk energy band structure of the trilayer film along the high symmetry points. The bands far from the Fermi energy are twofold degenerate, and the bands near the Fermi energy are not degenerate. (b) Locations of the four Weyl points in the first Brillouin zone, with Weyl points indicated by cyan. (c) Energy band structure of nanoribbons. The solid red and dashed blue lines indicate gapless helical edge states. (d1) and (d2) denote the probability distributions of the edge states for $E = 0.2$ in A and B of (c), respectively. The blue and red edge states on the same boundary propagate in opposite directions, hence the overall helical edge state. The parameters are chosen as $v_F = 1$, $m_1 = 1$, $m_0 = -2$, and $J = 0$.

point groups. We emphasize that our model should be viewed as a symmetry-guided minimal construction, rather than a material-specific tight-binding model. A full multiorbital derivation from first principles would indeed require a detailed treatment of spin, orbital, and crystal symmetry couplings, which could be pursued in future studies. Nonetheless, our simplified $\tilde{H}_{AM}$ captures the key ingredients for realizing momentum-dependent mirror-symmetric exchange fields that gap out edges and generate higher-order topology.

Throughout this work, we adopt dimensionless units by setting the lattice constant $a = 1$ and Planck constant $\hbar = 1$, following standard practice in tight-binding lattice models. Within this convention, parameters such as the Fermi velocity $v_F$, interlayer coupling strength $m$, and altermagnetism $J$ possess the same dimensional form, although they originate from distinct physical mechanisms. For reference, the Fermi velocity $v_F$ in $Bi_2Se_3$ thin films typically lies in the range of $(2-5) \times 10^5$ m/s [68]. The interlayer coupling $m$ corresponds to an energy scale of several to tens of meV depending on the film thickness [69]. The exchange coupling $J$ induced by proximity to a $d$-wave altermagnet is estimated to be a few meV up to about 10 meV [33]. Based on these experimental es-

timates, we confirm that the parameters chosen in our model are physically reasonable.

## III. RESULTS AND DISCUSSION

### A. Weyl semimetal with helical edge state

Let us first examine the system without the altermagnet ($J = 0$). The bulk band structure of the trilayer films along the high-symmetry points is shown in Fig. 2(a). One can see that the bulk bands far from the Fermi energy are twofold degenerate, while those bulk bands near the Fermi energy are non-degenerate and close at the high-symmetry points. It indicates that the Weyl points in the 2D system are fixed at these high-symmetry points, and are not cleaved from Dirac point because of the lack of a spatial inversion counterpart. The positions of the Weyl points (highlighted in cyan) within the Brillouin zone are shown in Fig. 2(b).

To further investigate the edge states, we calculate the energy spectrum of nanoribbons as a function of $k_x$, with the $y$-direction subjected to open boundary conditions ($N_y = 50a$), where $a$ denotes the lattice constant. Interestingly the 2D Weyl semimetal exhibits edge states, as illustrated by the red solid and blue dashed lines in Fig. 2(c). To characterize these states more precisely, we analyze the wave function distribution corresponding to the edge states at energy $E = 0.2$, as shown in Fig. 2(d1) and Fig. 2(d2). We observe that the two edge states at point $A$ are localized at opposite ends of the nanoribbon. Similarly, the edge states at point $B$ are also distributed at opposite ends, but their spatial distribution is reversed compared to those at point $A$. This spatial distribution confirms that these are helical edge states. Thus, the above analysis establishes that the trilayer film system is a 2D Weyl semimetal with helical edge states.

To elucidate the physical origin of the Weyl points and helical edge states, we examine the symmetry of the bulk Hamiltonian $H(\boldsymbol{k})$. In the absence of altermagnet, the Hamiltonian $H(\boldsymbol{k})$ retains invariance under the symmetry $\mathcal{M}$, where

$$\mathcal{M} = \frac{\sqrt{2}}{2} \begin{bmatrix} 0 & s_z & 0 \\ s_z & 0 & s_z \\ 0 & s_z & 0 \end{bmatrix}. \qquad (4)$$

We note that the mirror-like operator $\mathcal{M}$ is a Hermitian matrix that does not represent a conventional physical symmetry. Instead, it defines an algebraic symmetry of the lattice model, satisfying the commutation relation $[H(\boldsymbol{k}), \mathcal{M}] = 0$. This commuting property allows $H(\boldsymbol{k})$ and $\mathcal{M}$ to be simultaneously diagonalized. In particular, symmetric trilayer van der Waals heterostructures naturally provide the structural conditions for realizing the algebraic structure encoded by $\mathcal{M}$: the two outer layers can be made symmetry-related by stacking design, while the interlayer hybridization can be tuned via stacking ge-

ometry, spacer layers, or electrostatic gating. Such tunability enables the effective Hamiltonian to acquire the same block structure generated by $\mathcal{M}$, even though $\mathcal{M}$ is not a conventional mirror operation. In this basis, the eigenvectors of $\mathcal{M}$ form a unitary matrix $U$ that can be used to transform $H(\boldsymbol{k})$ into a block-diagonal form:

$$H'(\boldsymbol{k}) = U^\dagger H(\boldsymbol{k})U,$$

where each block corresponds to a subspace labeled by a distinct eigenvalue of $\mathcal{M}$. This decomposition enables the topological properties to be analyzed separately within each symmetry-respecting subspace. It is straightforward to find that $\mathcal{M}$ has three eigenvalues: $-1$, $0$, and $+1$. The corresponding eigenvectors form a unitary matrix $U$, which can be explicitly written as

$$U = \begin{pmatrix} 0 & \frac{1}{2} & \frac{\sqrt{2}}{2} & 0 & \frac{1}{2} & 0 \\ -\frac{1}{2} & 0 & 0 & -\frac{\sqrt{2}}{2} & 0 & -\frac{1}{2} \\ 0 & -\frac{\sqrt{2}}{2} & 0 & 0 & \frac{\sqrt{2}}{2} & 0 \\ -\frac{\sqrt{2}}{2} & 0 & 0 & 0 & \frac{\sqrt{2}}{2} & 0 \\ 0 & \frac{1}{2} & -\frac{\sqrt{2}}{2} & 0 & \frac{1}{2} & 0 \\ -\frac{1}{2} & 0 & 0 & \frac{\sqrt{2}}{2} & 0 & -\frac{1}{2} \end{pmatrix}.$$

Here, the columns of $U$ correspond respectively to eigenvalues $-1$, $-1$, $0$, $0$, $+1$, and $+1$. In the subspaces associated with three eigenvalues, $H(k)$ can be decoupled into three independent parts:

$$H_{\pm 1} = -v_F \sin k_y \tau_x \pm v_F \sin k_x \tau_y + \sqrt{2}m(\boldsymbol{k})\tau_z,$$
$$H_0 = -v_F \sin k_y \tau_x + v_F \sin k_x \tau_y \equiv \mathbf{d}_0(k) \cdot \tau, \quad (5)$$

where $\tau = (\tau_x, \tau_y, \tau_z)$ are the Pauli matrix. Each of $H_{-1}, H_1$, and $H_0$ exhibits chiral symmetry, and the chiral symmetry operators are denoted by $(\tau_x - \tau_y)/\sqrt{2}$, $(\tau_x + \tau_y)/\sqrt{2}$ and $\tau_z$ respectively. Notably, $H_0$ is zero at the four high symmetry points $\Gamma(0,0), X_1(\pi,0), X_2(\pi,0)$, and $M(\pi,\pi)$, which correspond to the four Weyl points, as shown in Fig. 2(b). To determine the nature of the band touching, we expand $H_0$ around each point by writing $\mathbf{k} = \mathbf{k}_W + \mathbf{q}$ with $|\mathbf{q}| \ll 1$. This yields

$$H_0^{(W)}(\mathbf{q}) \simeq v_F \left( \alpha_W q_x \tau_y + \beta_W q_y \tau_x \right), \quad (6)$$

where $(\alpha_W, \beta_W) = (+1, -1), (-1, -1), (+1, +1), (-1, +1)$ for $W = \Gamma, X_1, X_2, M$, respectively. The corresponding dispersion

$$E_\pm^{(W)}(\mathbf{q}) = \pm v_F \sqrt{q_x^2 + q_y^2}, \quad (7)$$

is linear in $\mathbf{q}$, demonstrating that all four degeneracies are genuine linear Weyl crossings pinned to the high-symmetry points. We further investigate the topological charge associated with each Weyl point in the subspace Hamiltonian $H_0$. The winding number for each Weyl point is calculated using the following expression [1]:

$$v_W = \frac{i}{2\pi} \int_C dk \operatorname{Tr}[q^{-1}(\boldsymbol{k})\partial_k q(\boldsymbol{k})],$$
$$q(\boldsymbol{k}) = -v_F(\sin k_y + i \sin k_x), \quad (8)$$

where $W = \Gamma, X_1, X_2, M$ and $C$ denotes an enclosing loop around a Weyl point in the 2D Brillouin zone. It follows that $v_\Gamma = v_M = -v_{X_1} = -v_{X_2} = 1$, corresponding to the four Weyl points. The total topological charge of the two Weyl points along the $k_x = 0$ and $k_x = \pi$ projections is zero, i.e., $v_\Gamma + v_{X_2} = 0$ and $v_M + v_{X_1} = 0$. As a result, the energy band structure of the nanoribbons, shown in Fig. 2(c), does not exhibit one-dimensional Fermi arcs connecting the Weyl points. It is worth noting that quasi-flat one-dimensional Fermi arc edge states emerge, connecting Weyl points of opposite chirality, as long as the nanoribbon is oriented away from the $k_x$ or $k_y$ directions.

The Chern numbers of bulk gapped $H_1$ and $H_{-1}$ can be calculated using the following expression [70–72]

$$\mathcal{C} = \frac{1}{2\pi} \sum_n \int_{BZ} d^2\mathbf{k}\Omega_n, \quad (9)$$

where $\mathbf{k} = (k_x, k_y)$ and $\Omega_n$ is the momentum-space Berry curvature for the $n$th band [73–75]

$$\Omega_n(\mathbf{k}) = -\sum_{m \neq n} \frac{2\operatorname{Im}\langle\psi_{n\mathbf{k}}|v_x|\psi_{m\mathbf{k}}\rangle\langle\psi_{m\mathbf{k}}|v_y|\psi_{n\mathbf{k}}\rangle}{(\omega_m - \omega_n)^2}. \quad (10)$$

In this expression, the sum runs over all occupied bands below the bulk gap, where $\omega_n = E_n/\hbar$ represents the energy of the $n$-th band, and $v_{x,y}$ are the velocity operators. By integrating the Berry curvature, we calculate the Chern numbers and find $\mathcal{C}_{H_1} = -\mathcal{C}_{H_{-1}} = 1$, which corresponds to a pair of the opposite chiral edge states in open boundary, i.e., helical edge states. These integer Chern numbers remain numerically stable under Brillouin-zone mesh refinement, with their values unchanged within machine precision. These results indicate that the Hamiltonians of the three subspaces collectively describe a 2D Weyl semimetal with helical edge states.

**B. Corner states in Weyl semimetal**

We now introduce a $d$-wave altermagnet ($J = 0.6$) to demonstrate the emergence of corner states. The energy spectrum of the nanoribbon is shown in Fig. 3(a), with the helical edge states being gapped by the altermagnetism (indicated by the solid blue and dashed red lines). Notably, the momentum-dependent $d$-wave altermagnetic field vanishes along the $k_x = \pm k_y$ directions, preserving band degeneracy and yielding a quantized Berry phase of $\pi$ along these cuts. As a result, topologically protected edge states emerge in ribbons oriented along the [11] and [$\bar{1}$1] directions. Interestingly, the bulk bands remain unaffected. Despite the disappearance of the first-order edge states, we observe the emergence of second-order topological phases. To investigate corner states, we analyze the energy spectrum of finite-size nanoflakes, such as a rectangular nanoflake with dimensions $40a \times 40a$. As shown in Fig. 3(b), four zero-energy

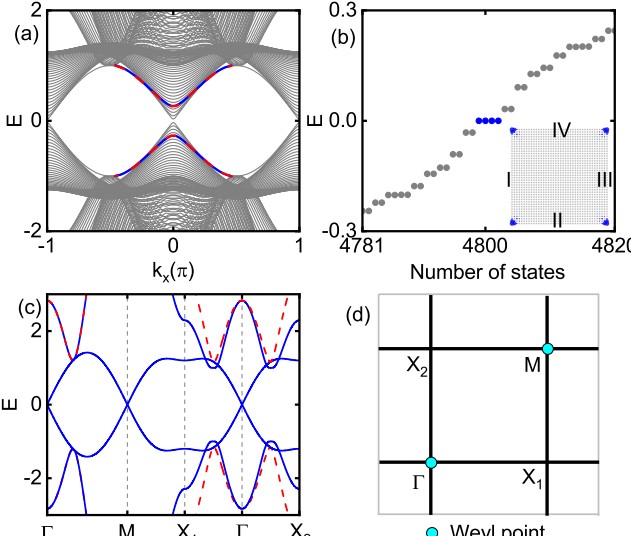

FIG. 3. (a) Band structures of nanoribbons of the topological film with $d$-wave altermagnet. The blue solid line and the red dashed lines indicate the gapped helical edge states. (b) Energy levels of the rectangular-shaped nanoflake. Blue dots correspond to corner states. The wave function probability distribution of the corner states is shown in the inset (blue density dots). (c) Bulk energy spectrum along the high symmetry points. (d) Locations of the two Weyl points in the first Brillouin zone, with Wely points indicated by cyan. The altermagnet is set to $J = 0.6$ and the other parameters are the same as in Fig. 2.

states appear at the Fermi level (highlighted by blue dots) within the continuum spectrum. The inset shows the wave-function probabilities of these corner states at half-filling, revealing that the electronic charge is localized at the corners of the nanoflake. The number and spatial localization of the corner modes remain unchanged for different nanoflake sizes, indicating that they are not finite-size artifacts.

The energy bands shown in Fig. 3(a) confirm that the bulk bands remain gapless. However, a key characteristic of a semimetal is the crossover of the bulk bands. We plot the bulk band structure of the topological film with $d$-wave altermagnet in Fig. 3(c). It is clear that while the Weyl points at $X_1$ and $X_2$ become gapped, the Weyl points at $\Gamma$ and $M$ persist. The positions of these remaining Weyl points in the first Brillouin zone are indicated by cyan markers in Fig. 3(d). The simultaneous presence of corner states in finite-size nanoflakes and the Weyl points in the bulk band directly signals the realization of 2D higher-order Weyl semimetals. Because the four corner modes are spatially separated, finite-size hybridization can generate a small splitting away from zero energy. For the system size employed in our calculations, however, this splitting is already negligibly small, and the corner states remain well isolated and strongly localized. We emphasize that the realization of the higher-order Weyl semimetal phase requires a uniform out-of-plane $d$-

wave altermagnetic exchange field applied to both the top and bottom layers. When the orientations differ, no corner states emerge, indicating that the second-order topological phase cannot be realized under such conditions.

In conventional Weyl semimetals, Fermi-arc states typically connect Weyl points along specific momentum-space directions. In contrast, in our system, the two Weyl points always carry a net topological charge of zero for any fixed $k_x$ or $k_y$, thereby preventing the formation of flat-band Fermi arcs along these directions. As a consequence, no hybridization occurs between Fermi-arc states and corner states along the $k_x$ and $k_y$-oriented edges. Furthermore, the vanishing density of states at the Weyl nodes ensures that bulk states do not hybridize with the zero-energy corner modes, even as the system size increases.

To understand the physical origin of the corner states and the persistence of Weyl points, we analyze the symmetry properties of the system. Although the $d$-wave altermagnet breaks time-reversal symmetry and the $\mathcal{M}$ symmetry, leading to gaps in both the helical edge states and the Weyl points at $X_1$ and $X_2$, the Hamiltonian along the high-symmetry directions $k_x = k_y$ and $k_x = -k_y$ retains $\mathcal{M}$ symmetry. Here, the symmetry operator $\mathcal{M}$ is a Hermitian matrix that commutes with the Hamiltonian along high-symmetry lines, i.e., $[\mathcal{M}, H(k_x, \pm k_x)] = 0$. We emphasize that $\mathcal{M}$ is not associated with any physical mirror reflection or rotational symmetry of the crystal lattice. Rather, it represents an algebraic symmetry emergent from the model's structure, which enables block-diagonalization of the Hamiltonian into symmetry-respecting subspaces and facilitates the analytical computation of topological invariants such as winding numbers. In particular, this allows $H(k_x, k_x)$ to be decomposed into three subspace Hamiltonians,

$$H_{\pm 1} = v_F \sin k_x (\tau_x \mp \tau_y) + \sqrt{2} m(k_x) \tau_z \equiv \mathbf{d}_{\pm 1}(k) \cdot \tau,$$
$$H_0 = v_F \sin k_x (\tau_x - \tau_y). \tag{11}$$

Clearly, the subspace Hamiltonians $H_{-1}$, $H_1$, and $H_0$ all exhibit chiral symmetry, and the chiral symmetry operators are denoted by $(\tau_x - \tau_y)/\sqrt{2}$ , $(\tau_x + \tau_y)/\sqrt{2}$ and $\tau_z$ respectively. The Hamiltonians $H_0$ has two gapless points in momentum space at $k_x = 0$ and $k_x = \pi$, corresponding to the Weyl points located at the $\Gamma$ and $M$ points. For the bulk gapped Hamiltonians $H_{-1}$ and $H_1$, we calculate their winding numbers using the following expression[76]:

$$v_{\pm 1} = \frac{1}{2\pi i} \int_{-\pi}^{\pi} dk_x \frac{d \ln h(k_x)}{dk_x}, \tag{12}$$

where $h(k_x)$ denotes the Hamiltonian $H_{\pm 1}$ and it is evaluated along the high symmetry line $k_x = \pm k_y$. The calculated winding numbers are $v_{\pm 1} = \pm 1$, which indicates the presence of two corner states in the 2D finite-size system, located at the endpoints of the $k_x = k_y$ path.

For the case of $k_x = -k_y$, the Hamiltonian $H(k_x, -k_x)$ can similarly be decomposed into three distinct components:

$$H_{\pm 1} = v_F \sin k_x(\tau_x \pm \tau_y) + \sqrt{2}m(\boldsymbol{k})\tau_z,$$
$$H_0 = v_F \sin k_x(\tau_x + \tau_y). \qquad (13)$$

The winding numbers $v_{\pm 1} = \mp 1$ are calculated for $H_{-1}$ and $H_1$, confirming the presence of two corner states at the endpoints of the $k_x = -k_y$ path. Therefore, the corner states are not trivial defect modes but intrinsic topological boundary states. Their emergence requires edge gap opening induced by a finite $d$-wave exchange field $J$ and is characterized by a quantized winding number within the symmetric subspaces. The absence of corner states at $J = 0$ and their appearance for $J > 0$ signify a topological phase transition from a first-order to a second-order Weyl semimetal, which may be probed experimentally via scanning tunneling microscopy (STM). This analysis demonstrates that the system is a 2D higher-order Weyl semimetal, characterized by bulk Weyl points and corner states. These findings are in excellent agreement with the numerical results presented in Fig. 3.

## C. Edge theory

In this subsection we derive an effective edge theory for the trilayer Hamiltonian and analytically clarify the origin of the corner zero modes in the higher-order Weyl phase. We start from the lattice model $H(\boldsymbol{k}) + H_{AM}$ and expand it around the $\Gamma$ point. Using $\sin k_{x,y} \simeq k_{x,y}$ and $\cos k_{x,y} \simeq 1 - k_{x,y}^2/2$, we obtain the continuum form

$$H(\boldsymbol{k}) \simeq v_F(k_y s_x - k_x s_y)\rho_z + \left[m_0 + m_1(k_x^2 + k_y^2)\right]s_0\rho_x$$
$$- \frac{J}{2}(k_x^2 - k_y^2)s_z P_{tb}, \qquad (14)$$

where $P_{tb} = \text{diag}(1,0,1)$ acts in the trilayer space and selects the top and bottom layers.

Diagonalizing the trilayer matrix $\rho_x$ yields three eigenvalues $0, \pm\sqrt{2}$ with orthonormal eigenvectors $\boldsymbol{u}_0, \boldsymbol{u}_-, \boldsymbol{u}_+$. In our model the helical edge states originate from the bands associated with $\boldsymbol{u}_\pm$, while the band corresponding to $\boldsymbol{u}_0$ does not participate in the edge modes and is neglected in the following. We therefore project onto the two-dimensional subspace spanned by $\boldsymbol{u}_\pm$. Introducing the $3 \times 2$ projection matrix $Q = (\boldsymbol{u}_+, \boldsymbol{u}_-)$ the effective layer operators become

$$\rho_x^{\text{eff}} = Q^\dagger \rho_x Q = \sqrt{2}\,\sigma_z, \qquad \rho_z^{\text{eff}} = Q^\dagger \rho_z Q = \sigma_x, \quad (15)$$

where $\sigma_{x,z}$ are Pauli matrices in the effective layer space. For later convenience we define

$$\tau_z' \equiv \sigma_x, \quad \tau_x' \equiv \sigma_z, \quad m_0' = \sqrt{2}\,m_0, \quad m_1' = \sqrt{2}\,m_1. \qquad (16)$$

Projecting the kinetic and interlayer-hybridization terms in Eq. (14) then gives the effective continuum Hamiltonian

$$H_0^{\text{eff}}(\boldsymbol{k}) = v_F(k_y s_x - k_x s_y)\,\tau_z' + \left[m_0' + m_1'(k_x^2 + k_y^2)\right]\tau_x', \qquad (17)$$

which reproduces the helical edge structure. The $d$-wave altermagnetic term is projected to

$$H_{\text{AM}}^{\text{eff}}(\boldsymbol{k}) \simeq -\frac{J_{\text{eff}}}{2}(k_x^2 - k_y^2)s_z\tau_z', \qquad (18)$$

where $J_{\text{eff}} \propto J$ is a renormalized exchange parameter determined by $Q^\dagger P_{tb}Q$. This effective term acts as a momentum-dependent Zeeman field in the edge subspace and is responsible for opening Dirac masses on the boundaries.

For ease of discussion, the four edges of a square nanoflake are labeled as I, II, III, and IV in counterclockwise order, as shown in Fig. 3(b), where edges I and III are parallel to the $y$ axis and edges II and IV are parallel to the $x$ axis. We first focus on edge I, which corresponds to a semi-infinite region $x \geq 0$ with the boundary at $x = 0$. Along this edge $k_y$ is a good quantum number, while $k_x \to -i\partial_x$. The effective Hamiltonian is conveniently decomposed as

$$H_{\text{eff}}(-i\partial_x, k_y) = H_0(-i\partial_x) + H_p(-i\partial_x, k_y), \qquad (19)$$

with

$$H_0(-i\partial_x) = iv_F\partial_x s_y\tau_z' + \left(m_0' - m_1'\partial_x^2\right)\tau_x', \qquad (20)$$

$$H_p(-i\partial_x, k_y) = v_F k_y s_x\tau_z' + \frac{J_{\text{eff}}}{2}\partial_x^2 s_z\tau_z'. \qquad (21)$$

Here we have used $k_x^2 - k_y^2 \simeq -\partial_x^2$ for modes localized near the edge and neglected the $k_y^2$–dependent corrections in the mass term, which only give subleading changes to the gap magnitude and do not affect the sign of the Dirac mass. The unperturbed part $H_0$ describes the helical edge modes of the pristine trilayer film, while $H_p$ encodes the dispersion along the edge and the $d$-wave altermagnetic perturbation.

To obtain the edge modes, we set $k_y = 0$ and $H_p = 0$ and solve the zero-energy equation $H_0(-i\partial_x)\psi(x) = 0$ on the half-line $x \geq 0$ with hard–wall boundary conditions $\psi(0) = \psi(+\infty) = 0$. This equation admits two linearly independent normalizable solutions of the form

$$\psi_\alpha(x) = f(x)\,\chi_\alpha, \qquad \alpha = 1,2, \qquad (22)$$

where $f(x)$ is an exponentially decaying function and the spinors $\chi_\alpha$ satisfy $\left(is_y\tau_z' + s_0\tau_x'\right)\chi_\alpha = 0$. These two states span the low-energy subspace of helical edge modes on edge I. In this two–dimensional subspace we introduce Pauli matrices $\eta_{x,y,z}$ acting on the edge degrees of freedom.

Treating $H_p$ as a perturbation, the effective edge Hamiltonian for edge I is obtained by projecting $H_p$ onto the subspace spanned by $\{\psi_1, \psi_2\}$,

$$(H_I)_{\alpha\beta}(k_y) = \int_0^\infty dx\, \psi_\alpha^\dagger(x) H_p(-i\partial_x, k_y)\psi_\beta(x). \quad (23)$$

The first term in Eq. (21) generates the kinetic term

$$\int_0^\infty dx\, \psi_\alpha^\dagger(x) v_F k_y s_x \tau_z' \psi_\beta(x) = v_F k_y (\eta_x)_{\alpha\beta}, \quad (24)$$

while the second term produces a Dirac mass,

$$\int_0^\infty dx\, \psi_\alpha^\dagger(x) \frac{J_{\text{eff}}}{2} \partial_x^2 s_z \tau_z' \psi_\beta(x) = m_I (\eta_z)_{\alpha\beta}, \quad (25)$$

with $m_I = C_I \frac{J_{\text{eff}} m_0'}{2m_1'}$, where $C_I > 0$ is a dimensionless constant determined by the detailed profile of $f(x)$ and does not affect the sign of the mass. Thus the low-energy dispersion on edge I is described by the one-dimensional Dirac Hamiltonian

$$H_I(k_y) = v_F k_y \eta_x + m_I \eta_z. \quad (26)$$

The sign of the Dirac mass $m_I$ is controlled by the sign of $J_{\text{eff}} m_0'$, while its magnitude is set by the altermagnetic exchange scale and the bulk gap parameters. In the absence of altermagnetism ($J_{\text{eff}} = 0$), edge I hosts a pair of gapless helical modes, consistent with the numerical spectra in Fig. 2(c); a finite $J_{\text{eff}}$ gaps these modes and generates a Dirac mass $m_I$ on edge I.

For the other edges (II, III, IV), similar analyses yield the effective Hamiltonians

$$\begin{aligned}
H_{II} &= -v_F k_x \eta_y - m_I \eta_z, \\
H_{III} &= -v_F k_y \eta_x + m_I \eta_z, \\
H_{IV} &= v_F k_x \eta_y - m_I \eta_z.
\end{aligned} \quad (27)$$

Thus the Dirac effective masses at the four edges I, II, III, and IV are given by $m_I$, $-m_I$, $m_I$, and $-m_I$, respectively. These mass terms indicate the formation of Dirac mass domain walls at all four corners of the rectangular nanoflake. According to the Jackiw-Rebbi mechanism [31, 77], such mass-domain-wall configurations host zero-energy bound states at the corners, which is fully consistent with the corner-localized states found in the numerical spectra in Fig. 3(b).

### D. Robustness against disorder

Finally, we examine the stability of the corner states against disorder in the higher-order Weyl phase. As discussed in Sec. III B, the zero-energy corner modes originate from a quantized winding number defined within the $\mathcal{M}$-symmetric subspaces of the effective Hamiltonian. Perturbations that preserve this subspace structure and

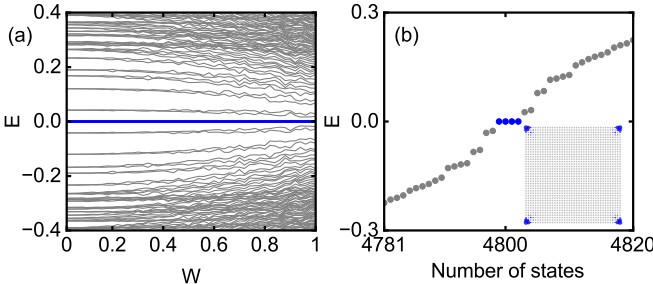

FIG. 4. (a) Energy spectrum of a $40a \times 40a$ nanoflake as a function of the strength $W$ of the random magnetic disorder $H_{\text{dis}}^M$. Gray lines denote bulk and edge states, while the four corner levels are highlighted in blue. The corner states remain pinned near zero energy over a wide range of $W$. (b) Eigenvalue spectrum near zero energy for a fixed disorder strength $W = 0.5$. Blue dots mark the four corner states, and the inset shows their wave-function probability distribution, localized at the four corners of the nanoflake.

do not close the corresponding gaps are therefore not expected to destroy the higher-order boundary modes.

To test this explicitly, we first consider magnetic disorder in the form of a random exchange field

$$H_{\text{dis}}^M = \sum_i \delta J_i\, c_i^\dagger \big(\mathbb{I}_{\text{layer}} \otimes s_z\big) c_i, \qquad \delta J_i \in [-W, W]. \quad (28)$$

This term preserves the overall spin quantization axis and the $\mathcal{M}$-subspace structure of the effective Hamiltonian. Figure 4(a) shows the evolution of the low-energy spectrum of a $40a \times 40a$ nanoflake as a function of the disorder strength $W$. Gray lines denote bulk and edge states, while the four in-gap levels are highlighted in blue. Over the whole range $0 \leq W \leq 1$, these levels remain pinned close to zero energy, whereas all other states form a broadened continuum. The corresponding eigenvalue spectrum near zero energy for $W = 0.5$ is shown in Fig. 4(b); the four corner states are marked by blue dots, and the inset confirms that their wave-function probability is strongly localized at the four corners of the nanoflake. These results demonstrate that the corner modes are robust against symmetry-preserving magnetic disorder.

In contrast, scalar onsite-potential disorder of the form $H_{\text{dis}}^V = \sum_i w_i c_i^\dagger c_i$, acts identically on all internal degrees of freedom, mixes different $\mathcal{M}$ subspaces, and strongly couples the corner modes to the gapless bulk continuum. As the disorder strength $W$ grows, the four in-gap levels broaden and eventually merge into the bulk spectrum, so that the corner states are no longer spectrally isolated. This behavior reflects a generic feature of gapless higher-order Weyl semimetals: corner modes are protected against symmetry-preserving perturbations, but not against arbitrary, symmetry-breaking scalar disorder.

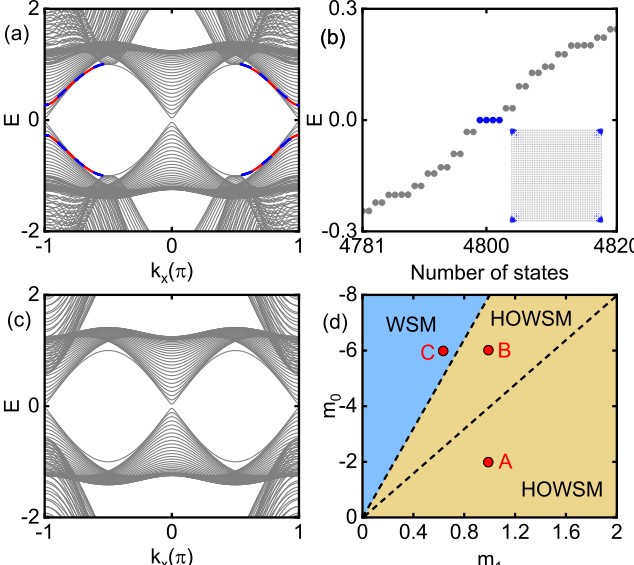

FIG. 5. (a) The energy bands correspond to the parameters $v_F = 1, m_0 = -6, m_1 = 1$, and $J = 0.6$. The blue solid lines and the red dashed lines indicate the gapped helical edge states. (b) Energy levels of the rectangular-shaped nanoflake. Blue dots correspond to corner states. The wave function probability distribution of the corner states is shown in the inset (blue density dots). (c) The energy bands correspond to the parameters $v_F = 1, m_0 = -6, m_1 = 0.7$, and $J = 0.6$. (d) Topological phase diagram of the trilayer film as functions of $m_1$ and $m_0$ with $v_F = 1$, and $J = 0.6$. The black dashed lines indicates the phase boundaries ($|m_0| = 8|m_1|$ and $|m_0| = 4|m_1|$). HOWSM corresponds to a higher-order Weyl semimetal phase with corner states, WSM for normal Weyl semimetal. Parameter point coordinates $(m_1, m_0) : A(1, -2)$, $B(1, -6)$, and $C(0.7, -6)$.

## E. Topological phase diagram

In the above analysis, we have chosen specific parameters to analyze the topological phases. In this section, we systematically explore the effect of varying parameters on the topological phase of the system. We first adjust the value of $m_0$, setting the system parameters to $v_F = 1, m_0 = -6, m_1 = 1$, and $J = 0.6$, i.e. changing $m_0 = -2$ in Figs. 2 and 3 to $m_0 = -6$. Interestingly, as shown in the energy band structure in Fig. 5(a), the gapped helical edge states now appear at $k_x = \pi$, which is different from the case shown in Fig. 3(a). To understand the topological properties of the gapped edge states, we plot the energy levels of the rectangular-shaped nanoflake in Fig. 5(b). The inset displays the wavefunction distributions of the four zero-energy states (highlighted in blue), uniformly distributed over the four corners, indicating a two-dimensional higher-order topological phase. Although the system remains in the higher-order Weyl semimetal phase both before and after adjusting $m_0$, the edge states shift from $k_x = 0$ to $k_x = \pi$), suggesting that a topological phase transition has occurred. We next ad-

just $m_1$ to analyze the system in more detail, with the system parameters set to $v_F = 1, m_0 = -6, m_1 = 0.7$, and $J = 0.6$. The energy band structure of the nanoribbon is shown in Fig. 5(c), where the helical edge state disappears, and the system behaves as a normal Weyl semimetal phase.

To clearly understand and determine the phase boundaries, we analyze the Hamiltonian of the symmetric subspace in Eq. (13). It is clear that no matter how $m_1$ and $m_0$ vary, a bulk band of $H_0$ always closes, giving rise to two Weyl points in the trilayer system. The bulk band closure (or phase boundary) of $H_{\pm 1}$ corresponds to $m_0 m_1 = 0$ or $|m_0| = 8|m_1|$ or $|m_0| = 4|m_1|$, dividing the phase diagram in Fig. 5(d) into three distinct regions. Using Eq. (12), we calculate the winding number of $H_{\pm 1}$ and find that the winding number $v_{\pm 1} = \mp 1$ while $m_0 m_1 < 0$ and $|m_0| < 8|m_1|$ (excluding $|m_0| = 4|m_1|$), while $v_{\pm 1} = 0$ in all other cases. Therefore, the yellow region corresponds to the higher-order Weyl semimetal phase, and the blue region corresponds to the normal Weyl semimetal phase. This is consistent with the numerical results. The parameter points $A$, $B$, and $C$ correspond to the energy band structures in Figs. 3(a), 5(a), and 5(c), respectively. Since the $d$-wave altermagnetic term enters $H_{\pm 1}$ through $\Delta(\boldsymbol{k}) = \cos k_x - \cos k_y$, which vanishes at the bulk gap-closing momenta, it does not affect the mass term determining the phase boundaries, so the lines $|m_0| = 4|m_1|$ and $|m_0| = 8|m_1|$ in Fig. 5(d) remain independent of $J$ for moderate exchange fields such as $J = 0.6$.

## IV. CONCLUSIONS

We have demonstrated that an out-of-plane $d$-wave altermagnet can transform a trilayer topological film into a 2D higher-order topological Weyl semimetal, featuring corner states. Specifically, the pristine trilayer topological film functions as a 2D Weyl semimetal, exhibiting helical edge states and four bulk Weyl points. The systems's topological properties are characterized by the Chern number and the winding numbers of symmetric subspaces. Introducing an out-of-plane $d$-wave altermagnet on the top and bottom layers induces a gap in the helical edge states while preserving two bulk Weyl points. We have further analyzed the energy levels of a rectangular nanoflake, which confirms the existence of corner states. The emergence of these corner states is attributed to the nonzero winding number along the symmetric subspace of the high-symmetry line $k_x = \pm k_y$. Finally, we provide a complete second-order topological phase diagram. Our findings provide compelling insights into the realization of pioneering 2D higher-order Weyl semimetals.

We also point out that embedding the trilayer film between two layers of an altermagnetic material, such as MnTe, which has been proposed as a realistic altermagnet with appropriate symmetry and lattice compat-

ibility [78], offers a feasible route for realizing the proposed higher-order Weyl semimetal phase. Experimentally, charged topological corner states can be detected using STM [79]. When the STM probe is at the energy of the other states, a broadening peak can be observed. After the corner states are induced by altermagnet, sharp peaks can be observed at the corners of the sample with STM. Therefore, observing changes in the energy spectrum peaks with STM can be a strong evidence for the existence of corner states. Beyond STM, the bound corner modes are also expected to leave distinct fingerprints in transport. In particular, they can generate sharp resonant features in conductance as a function of gate voltage and enhance nonlocal resistance between corner contacts in multi-terminal geometries. We further point out that several experimentally accessible perturbations can, in principle, distinguish the proposed topological corner states from trivial localized modes. First, the altermagnetic exchange field $J$ is essential for generating the corner states, and varying its magnitude or reversing its sign, e.g., via magnetic reorientation, can drive a topological phase transition. Second, the corner modes are protected by crystalline mirror symmetry. Applying uniaxial strain or introducing asymmetry between the top and bottom layers can selectively break this symmetry and remove the corner states, while trivial defect states remain unaffected. Third, STM-based mapping of spatial charge density provides a direct method for identifying symmetry-related localization at the corners, offering an additional route to distinguish the topological origin. These knobs make our model not only theoretically interesting but also experimentally testable in van der Waals heterostructures and magnetic film platforms. Beyond these experimental probes, the key ingredients required in our symmetry guided model are known to occur in magnetic topological insulator heterostructures. First principles studies of $MnTe/Bi_2(Se,Te)_3/MnTe$ stacks have demonstrated strong proximity induced exchange coupling and related topological phases [80], while ARPES measurements on $MnSe/Bi_2Se_3$ interfaces have observed a sizable exchange driven Dirac gap [81]. These works indicate that the essential building blocks of our mechanism are experimentally accessible in van der Waals magnetic heterostructures. A concrete material realization of the higher order Weyl semimetal proposed here requires a dedicated ab initio study, which lies beyond the scope of the present work and will be pursued in future investigations.

## ACKNOWLEDGEMENTS

This work was financially supported by the National Natural Science Foundation of China (Grants No. 12074097, No. 12374034, and No. 11921005), Natural Science Foundation of Hebei Province (Grant No. A2024205025), the National Key R and D Program of China (Grant No. 2024YFA1409002), and the Innovation Program for Quantum Science and Technology (Grant No. 2021ZD0302403).

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
