# Peer review of "Two-dimensional higher-order Weyl semimetals"

_SciPost Physics_

## Round 1 · Referee Report · Anonymous (Referee 1) · 2025-11-4

Report

This manuscript proposes a theoretical model for realizing 2D higher-order Weyl semimetals (HOWSMs) via a trilayer topological film (inspired by Bi2Se3) coupled to a d-wave altermagnet. The pristine film exhibits 2D Weyl semimetals with helical edge states and four Weyl points. Introducing z-directed d-wave altermagnetism gaps the edges, preserves two Weyl points, and induces corner states, characterized by subspace winding numbers. A phase diagram delineates HOWSM and standard Weyl semimetal phases.

The work is novel in extending higher-order topology to 2D gapless systems, leveraging altermagnets, a recently proposed magnetic order, for momentum-selective edge gapping while preserving bulk Weyl nodes. This "gapless-bulk, gapped-boundary" paradigm is intriguing and could inspire hybrid topological-magnetism studies. The subspace decomposition via algebraic symmetry M enables analytical invariants (Chern/winding numbers), providing clear topological classification. Numerical results (band structures, nanoflakes) consistently support claims.

A key weakness is the phenomenological nature: the Hamiltonian, while symmetry-guided, lacks tight-binding derivation from realistic materials, limiting experimental relevance (e.g., Bi2Se3 trilayers may not match exactly). The M symmetry is artificial (not crystal-derived), potentially over-idealized; its realizability in van der Waals structures needs justification. Logic is mostly consistent, but inconsistencies arise: Weyl points are claimed at high-symmetry points, yet 2D Weyl nodes typically require accidental crossings—verify if dispersions are truly linear (Fig. 2(a) suggests quadratic touches at some points). Altermagnet justification is perturbative but assumes uniform J; real proximity effects may vary. Phase diagram boundaries (|m0|=4/8|m1|) derive from subspace closures, but exclude J dependence—small J is stated not to affect it, yet Fig. 3 uses J=0.6, risking oversight for larger J. No discussion of disorder/interaction effects on corner states.

Advice for Improvement: Link model to specific materials (e.g., MnTe/Bi2Se3 interfaces) with ab initio estimates for parameters. Analytically derive corner state wavefunctions or use effective models for edges. Expand experimental probes: beyond STM, suggest transport (e.g., nonlocal resistance for corners) or ARPES for Weyl nodes. Address finite-size effects on hybridization. Revise for clarity: define "higher-order" explicitly in 2D context; add error bars or convergence checks for numerics.

The concept is innovative and timely, but requires stronger material ties and error checks for publication. I recommend major revisions.

Recommendation

Ask for major revision

  • validity: high
  • significance: high
  • originality: top
  • clarity: high
  • formatting: excellent
  • grammar: excellent

Author:  Yingtao Zhang  on 2025-12-05  [id 6107]

(in reply to Report 1 on 2025-11-04)
Category:
answer to question

We thank the referee for carefully reading our manuscript and the valuable report. We provide our response to the questions raised by the referee in the attached file.

Attachment:

Reply_to_Referee.pdf

---

## Round 2 · Referee Report · Anonymous (Referee 2) · 2025-12-25

Report

This theoretical study proposes an innovative scheme to realize a two-dimensional higher-order Weyl semimetal by coupling a trilayer topological film with a d-wave altermagnet. The pristine trilayer system exhibits helical edge states and hosts four bulk Weyl points at high-symmetry points. Introducing an out-of-plane d-wave altermagnetic exchange selectively gaps the helical edges while preserving two Weyl points, leading to the emergence of topological corner states within the edge gap. The corner modes are characterized by quantized winding numbers in symmetric subspaces and remain robust against symmetry-preserving disorder. The results are reliable, novel, and detailed. I recommend publication in SciPost after addressing the following minor points:
1. It can be observed that there is still a small energy gap in the band structures (bulk bands) of nanoribbons as shown in Fig. 2(c) and Fig. 3(a). Is this due to the insufficient size of the system? The authors should address this point.
2. Higher-order topological phases are typically studied in systems with a full global band gap. If the bulk gap closes completely in the thermodynamic limit, can the corner states still be clearly distinguished? Is it possible for them to hybridize with bulk states? It would be helpful to include a discussion explaining why the corner states can remain identifiable in such a scenario.

Attachment

Recommendation

Publish (surpasses expectations and criteria for this Journal; among top 10%)

---

## Round 2 · Author Response

Dear Editor,

We thank you for forwarding the referee’s report on our manuscript “Two-dimensional higher-order Weyl semimetals” (scipost_202507_00076v1). We appreciate the referee’s detailed and constructive comments and have revised the manuscript accordingly.

The referee provided an overall positive and encouraging assessment. He/She rated the validity and significance of the work as high, the originality as top, and the clarity, formatting, and grammar as high or excellent. The report emphasizes the novelty of realizing a two-dimensional gapless phase in which bulk Weyl points coexist with higher-order corner states, and the usefulness of the algebraic subspace decomposition for defining analytical Chern and winding numbers supported by numerical band structures and nanoflake spectra.

In addition, the referee also offered several suggestions to further strengthen the manuscript, calling for clarification and extension at both the conceptual level and in terms of experimental relevance and numerical robustness. In the revised manuscript, we have implemented targeted changes along these lines: we clarify the scope of our symmetry-guided minimal model and the role of the algebraic symmetry M, provide a sharper characterization of the Weyl nodes and phase diagram through explicit low-energy and subspace analyses, and add an edge-theory and disorder study that underpins the origin and robustness of the corner states. We also expand the discussion of higher-order topology in two dimensions, finite-size effects, possible experimental probes, and numerical convergence. All new or modified text is highlighted in blue in the revised version.

We hereby resubmit the revised manuscript together with a detailed point-by-point response to the referee’s comments. We believe that the concerns raised in the report have been satisfactorily addressed and that the main results are now presented in a clearer and more comprehensive manner. We hope that the revised version will be suitable for publication in SciPost Physics.

Yours sincerely,
Lizhou Liu, Qing-Feng Sun, and Ying-Tao Zhang

---

## Round 2 · List of Changes

List of Modifications

1) We have added a new Fig. 4, and the original Fig. 4 has been renumbered as Fig. 5.
2) We have added a new reference,
[77] R. Jackiw and C. Rebbi, Solitons with fermion number 1/2, Phys. Rev. D 13, 3398 (1976).
3) We have clarified the definition of 2D higher-order topology by stating that gapped edges and in-gap corner states characterize a second-order phase (see page 1, column 1, paragraph 1, lines 7–13.)
4) We have clarified the physical meaning and realizability of the algebraic symmetry M in symmetric trilayer vdW structures (see page 3, column 2, last paragraph, last lines 1-6 and page 4, column 1, lines 1-4).
5) We have demonstrated the linear nature of all Weyl points by adding an explicit low-energy expansion around the four high-symmetry momenta (see page 4, column 1, paragraph 1, last lines 4-14).
6) We have added explicit statements confirming the numerical convergence of the Chern numbers and the size-independence of the corner modes (see page 4, column 2, paragraph 2, last lines 3-6; page 5, column 1, paragraph 1, last lines 1-4).
7) We have added a clarification on finite-size hybridization, noting that the corner-state splitting is exponentially small for the system sizes used (see page 5, column 1, paragraph 2, last lines 3-8).
8) We have added a full edge theory derivation that analytically explains the origin of the corner zero modes and shows how alternating Dirac masses arise on the four edges (see page 6-7, Sec. III C Edge theory).
9) We have added a subsection on robustness against disorder, showing that magnetic disorder preserves the corner modes while scalar disorder destroys them (see page 7, Sec. III D Robustness against disorder).
10) We have explained why the gap-closing conditions, and thus the phase diagram, remain independent of J for altermagnetism (see page 8, column 2, paragraph 2, last lines 1-7).
11) We have expanded the discussion of experimental probes and tunability of the corner modes (see page 9, column 1, paragraph 1, lines 10-15).
12) We added a concise materials motivation in the Conclusions to highlight that MnTe/Bi₂(Se,Te)₃/MnTe and MnSe/Bi₂Se₃ heterostructures exhibit proximity induced exchange coupling and related topological phases, supporting the feasibility of the ingredients used in our model (see page 9, column 2, paragraph 1). And, We added two new references associated with this materials motivation:
[80] N. Pournaghavi et al., Phys. Rev. B 103, 195308 (2021).
[81] A. V. Matetskiy et al., Appl. Phys. Lett. 107, 091604 (2015).

---

## Editorial Decision

in_refereeing